# Habits and perceptions regarding open science by researchers from Spanish institutions

**Candela Ollé**[1]*, **Alexandre López-Borrull**[1], **Remedios Melero**[2], **Juan-José Boté-Vericad**[3], **Josep-Manuel Rodríguez-Gairín**[3], **Ernest Abadal**[3]

**1** Professor in the Faculty of Information and Communication Sciences at Universitat Oberta de Catalunya, Barcelona, Spain, **2** Institute of Agrochemistry and Food Technology, of Consejo Superior de Investigaciones Científicas (CSIC), Valencia, Spain, **3** Departament de Biblioteconomia, Documentació i Comunicació Audiovisual & Centre de Recerca en Informació, Comunicació i Cultura, Universitat de Barcelona, Barcelona, Spain

* collec@uoc.edu

**Data Availability Statement:** All relevant data for this study are available in the CSUC Dataverse repository (https://doi.org/10.34810/data690).

## Abstract

The article describes the results of the online survey on open science (OS) carried out on researchers affiliated with universities and Spanish research centres and focused on open access to scientific publications, the publication process, the management of research data and the review of open articles. The main objective was to identify the perception and habits of researchers with regard to practices closely linked to open science and the scientific value added is that offers an in-depth picture of researchers as one of the main actors to whom this transformation and implementation of open science will fall. It focuses on the different aspects of OS: open access, open data, publication process and open review in order to identify habits and perceptions. This is to make possible an implementation of the OS movement. The survey was carried out among researchers who had published in the years 2020–2021, according to data obtained from WoS. It was emailed to a total of 8,188 researchers and obtained a total of 666 responses, of which 554 were complete, the rest being forms with some questions unanswered. The main results showed that open access still requires the diffusion of practices and services provided by the institution, as well as training (library or equivalent service) and institutional support from the competent authorities (vice rectors or equivalent) in specific aspects such as data management. In the case of data, around 50% of respondents stated they had stored data in a repository, and of all the options, the most frequently given was that of an institutional repository, followed by a discipline repository. Among the main reasons for doing this, we found transparency, visibility of data and the ability to validate results. For those who stated they had never stored data, the most frequent reasons for not having done so were privacy and confidentiality, the lack of a mandated data policy or a lack of knowledge of how to do it. In terms of open peer review, participants mentioned a certain reticence to the opening of evaluations due to potential conflicts of interest that may arise or because lower-quality content might be accepted in order to avoid conflicts. In addition, the hierarchical structure of senior researcher versus junior researcher might affect reviews. The main conclusions indicate a need for persuasion of OA

**Funding:** This study is part of the project "Open science in Spain: A global approach to assess its degree of implementation" (RTI2018-094360-B-I00) financed by the State Plan for Scientific and Technical Research and Innovation, of the Ministry of Science, Innovation and Universities of Spain. The funders had no role in study design, data collection and analysis, decision to publish, or preparation of the manuscript.

**Competing interests:** The authors have declared that no competing interests exist.

to take place; APCs are an economic barrier rather than the main criterion for journal selection; OPR practices may seem innovative and emerging; scientific and evaluation policies seem to have a clear effect on the behaviour of researchers; researchers state that they share research data more for reasons of persuasion than out of obligation. Researchers do question the pathways or difficulties that may arise on a day-to-day basis and seem aware that we are undergoing change, where academic evaluation or policies related to open science, its implementation and habits among researchers may change. In this sense, more and better support is needed on the part of institutions and faculty support services.

## 1. Introduction

UNESCO defines science as "an inclusive construct that combines various movements and practices aiming to make multilingual scientific knowledge openly available, accessible and reusable for everyone, to increase scientific collaborations and sharing of information for the benefits of science and society, and to open the processes of scientific knowledge creation, evaluation and communication to societal actors beyond the traditional scientific community" [1]. This concept involves an holistic view of the entire life cycle of research, and a challenge in terms of how to develop an idea of science of which openness is a relevant characteristic. The definition of open access to publications, a pillar of open science, has its origin in the Budapest Declaration [2] of 2002, which highlighted the need to open scientific output and eliminate barriers to its access. Twenty years later, there is still a way to go, particularly in terms of advancement in the conception of knowledge as a common good. Institutional and governmental support, and that of agencies funding research, has played an important role in advocating for open science, both in the establishment of policies and in the funding of infrastructure that supports the development of open science.

In this sense, in the case of Europe, the European Commission has in recent years backed this new reality, as laid out several years ago in its report "'Science 2.0': Science in transition" [3]. Many of the agencies that make up Science Europe have committed to supporting the principles of Plan S, to requiring projects funded by them have open access to publications with the considerations they deem appropriate [4].

The evaluation of scientific activity based not only on quantitative aspects but on qualitative aspects and which incorporates open science practices is another of the fronts on which Europe is advancing [5]. In this sense, the COARA (Coalition for advancing the research assessment) coalition, which joins together over 350 institutions in more than 40 countries, agreed in July 2022 on the document *The Agreement on Reforming Research Assessment* [6], which establishes new principles for the evaluation and acknowledgement of research activity.

Just as open access to scientific articles had a clear dissemination and an erratic beginning between goodwill and obligation, in the case of research data, requirements that the data comply with FAIR [7] principles have been developed and highlighted by agencies funding research projects, including non-governmental agencies such as the Wellcome Trust and the European Commission itself, which has advocated for open access to publications since the Seventh Framework Programme and to research data since the Framework Programme H2020 up to the current Horizon Europe. No one is casting doubt on the advantages of open access to results derived from research, but there are still barriers to overcome in terms of the participation of researchers and their reticence toward sharing their data. The COVID-19 pandemic highlighted the way in which collaborative and shared scientific communication accelerated advances in the search for treatments for the virus, but there is still ground to be gained

with regard to the participation and acknowledgement of researchers who adopt open science practices [8, 9].

Spain is an outstanding country in scientific policy and output and has incorporated into its legislation aspects relating to open science, although none of its public funding agencies has undersigned Plan S. In the Spanish strategy for science and innovation for the period 2021–2027, open science forms part of its objectives and fosters, in consonance with EU directives, open access to the results of research, allowing data to be accessible, interoperable and reusable. The new law of science, passed on 5th September 2022 (act 17/2022) [10], in its article 37, indicates that research staff in the public sector whose research activity is primarily financed through public funds and who opt to disseminate their research results in scientific publications must store a copy of the final accepted version for publication and the data associated with it in open-access institutional or specialist repositories at the time of publication.

In addition, the active debate on how to appropriately evaluate researchers and research groups has been promoted by the academic world [5–11], and evaluation agencies themselves (ANECA, in Spain) display an active interest in reflecting on said aspects.

In the case of Spain, scientific literature compiles works in relation to open access at a general level [12], or in the case of certain disciplines, such as that of philosophy [13]. There are also several studies relating to the management of research data [14, 15] or to scientific policy [16]. A spotlight has also been shone on payment for Article Processing Charges [17, 18], and the motivations of senior researchers in relation to open science [19], and as far as we know, no other article has compiled the opinions of such a large collective of researchers with regard to open science and all its aspects, although Ruiz-Perez's [12] study is focused on researchers and open-access publishing. Finally, numerous studies have considered the ecosystem of Spanish scientific journals [20, 21].

So, it makes sense to explore which aspects researchers in Spain value regarding open access, the publication process, open data, and open review that might help to tackle the needs and concerns of researchers, and may assess new services that could be offered by administrations, libraries and general research support services. Our research group carried out qualitative studies that aided the formulation of the survey [21], as well as the study of other groups of interest, as in the case of scholarly journal editors [22], vice-rectors of research [23] and library services [24]. In this sense, the value of this article lies also in the fact that based on a state experience it is possible to discern recommendations in other environments. Moreover, prior qualitative research lays very solid foundations for going into depth on a collective that is key to the movement. Finally, we also pondered the effects of the COVID-19 pandemic as a reality that may have aided understanding of the importance of sharing research results in times of public health emergency.

The aims of this work were (1) uncover the perceptions of researchers regarding open access, open data, the publication process and open peer review on the Spanish scholarly publishing landscape and (2): what does this give us in research practices related to open science?. These questions will be addressed in the following sections. The article consists of, in addition to an introduction, the methodological section, results and discussion, discussion and conclusions, bibliographical references and annexes.

## 2. Methodology

In the phase prior to research, several interviews were carried out with researchers [21] whose results helped in preparing the questionnaire, which is the data collection technique for this article. The main objective is to explore first-hand and in depth the habits and perceptions of the research collective, since it is a key collective with a position that is central to the

movement. The questionnaire (S1 File) structure consisted of 4 blocks (knowledge of OA, publication process, research data and open review), plus one specifically on the effect of COVID-19, final considerations and sociodemographic questions (S1 File). The responses, depending on the question, could be single or multiple and others used a 5-point Likert scale (from least to most intensity, for example "strongly disagree" to "strongly agree") which were translated into a numerical scale from 1 to 5 for quantitative analysis and statistical treatment. At the end of each survey section, comments could be made on these which were also taken into account in the qualitative analysis and discussion of the results.

The database WOS was used to locate work published by Spanish authors in the period from 2020 to 2021. A script scanned the bibliographic references file line by line to find those beginning with "CORRESPONDENCE ADDRESS:" and containing "Spain" and extracted from them a word pattern compatible with an email address. Email addresses of researchers who had authored correspondence for the first 2000 articles and the last 2000 according to the chronological order of publication were automatically downloaded, and duplicates were removed. Before the sending, 10 teachers/researchers were selected to test the questionnaire. An anonymous survey had been sent to 8,188 people (3,317 women, 4,856 men, 12 undetermined and 4 generic email addresses- we obtained a total of 666 responses, of which 554 were complete) via email through the platform LimeSurvey (consent was given on the platform before starting the survey) in June 2021 and 2 reminders were then sent, ending in December 2021.

The descriptive and statistical analysis of questionnaire responses was done with the package SPSS v27 and the programme Excel 2016. For the quantitative analysis of questions that used the Likert scale, first the Cronbach alpha was calculated in order to verify the trustworthiness of the scale used. A factorial analysis was performed on these questions in order to reduce variables, and the Kaiser-Meyer-Olkin (KMO) test was used to verify that the factorial analysis was suitable in order to reduce the number of variables and replace them with their main components. Not all the data obtained were disseminated in this paper due to issues of interest, selection and limits [25].

Possible limitations identified in the study include potential biases by disciplines, the dynamic aspect of the debate and changes in legislation partway through. In addition, it would have to be repeated every so often in order to monitor any progress. Despite the limitations regarding the representativeness of the results of a qualitative study, as well as the WoS for obtaining contact emails, the research makes it possible to identify the habits and perceptions of researchers from Spanish institutions. Lastly, the treatment of monographs was explicitly omitted. The survey questions (S1 File) and the dataset Ollé [26] are available.

## 3. Results

### 3.1 Demographic data from the survey

In total, 911 accesses or logins to the online survey were counted; with blank responses eliminated, the total number was 666, of which 554 were complete responses and the remainder were forms containing some unanswered questions. In order to obtain a margin of error at a trust level of 95%, a total number of 385 was needed, so we can say that the responses obtained suffice for this margin of error. The number of men who answered the questionnaire was 328 (59.4%) and that of women 199 (36.1%). In terms of ages, the age range of 36–50 years was predominant at 45.8%, followed by older respondents (51–65) at 37.3% and young researchers (21–34) at 13.3%. The remaining percentage corresponded to older plus (>65).

50% of respondents belonged to public universities and 4.4% to private universities, 35% to a public research centre and 3% to private centres, and finally, 7.1% belonged to hospitals and

health centres. In terms of disciplines, researchers in health sciences prevailed with 44.4%, 14.7% in life sciences, 13.3% in experimental sciences, 9.1% in engineering, 8.7% in social sciences, 6.5% in mathematics and physical sciences, and finally 4% in art and the humanities.

With regard to the question on the source of research funding, 36% stated that it came from state funds, 21% from autonomous or local funding, 18% from European projects and other international agencies, and 13% stated that they had no external funding at the time of responding.

## 3.2 Knowledge of open science: Ground still to be covered

Although the meaning of open science is increasingly more prevalent in the academic population [27], there is work still to be done in order to consolidate its bases and the pillars on which it is founded.

In response to the question on whether the respondents knew the meaning of open science and its principles, 10.8% indicated that they had never heard of it, 27.1% said they had heard something about it but they had no idea of its implications, 25% said they agreed with the hypotheses and that the transition to open science needed to be faster, and 37% said they were aware of the change involved in its implementation in the academic system and that they did not believe it was prepared, for which reason the transition needed to be slower. It is noticeable that approximately one third of researchers stated they did not have a clear idea of the open science meaning despite the existence of mandates, directives and legislation about open access to publications and research data-sharing policies. One can deduct from this result the need for greater dissemination and explanation of the meaning of open science rather than simply the technical and formal implications in existing policy and legislation.

## 3.3 Open access to scientific publications

The block on open access consisted of 10 questions on the habitual practices of researchers in relation to publication, and in some more specific cases, relating to open-access publications.

Table 1 shows the frequency of publication in the last two years (2019–2020) and how many of these works were published with open access, whether in journals or open-access repositories.

According to this data, researchers stated they had openly published all their articles (~22%) or more than a half, and fewer than 10% had not openly published any of them. With regard to what version they used for archiving in repositories and where they tend to self-archive their work, the version of record (VOR) was the option with the most responses either in repositories or on academic networks where is permitted by the licence. Nearly 22% stated that they did not self-archive any version (Table 2).

**Table 1. Frequency of publication and open publication over the 2020–2021 period.**

| Question | Option | N | Percentage (%) |
|---|---|---|---|
| Articles published (2020–2021) | Fewer than 4 | 108 | 16.3 |
| | Between 4 and 7 | 154 | 23.2 |
| | More than 7 | 402 | 60.5 |
| Open publication, in journals and/or repositories | Half or more | 250 | 37.7 |
| | Fewer than half | 220 | 33.1 |
| | All | 144 | 21.7 |
| | None | 50 | 7.5 |

**Table 2. Publication version used for self-archiving and where it is stored.**

| Means of storage | N | Percentage (%) |
|---|---|---|
| VOR archiving in an open-access repository when licenses allowed | 266 | 40.2% |
| Archiving of the accepted version in an open-access repository when licenses allowed | 158 | 23.9% |
| Archiving of the preprint version in an open-access repository | 124 | 18.7% |
| VOR archiving on academic social networks (ResearchGate, Academia.edu) when licenses allowed or rights were owned | 257 | 38.8% |
| Archiving of the accepted version on social academic networks (ResearchGate, Academia.edu) where licenses allowed | 104 | 15.7% |
| Archiving of the preprint on social academic networks (ResearchGate, Academia.edu.) | 45 | 6.8% |
| Nothing (neither VOR nor the preprint in any place) | 145 | 21.9% |

On the other hand, when asked if open publication was a requirement of their funding agencies, 46% of respondents stated that it was, 42% that it was not, and 12% that they did not know, despite the fact that most funds received came from entities with policies on publication as indicated in the bases of their calls for publication, both national and European.

Those surveyed were asked for their reasons for publishing their work openly (656–663 responses) (Fig 1) as well as for their reasons for not doing so (Fig 2). The reasons for open publication with the highest scores in terms of being in agreement with them made reference to the increased visibility and, with it, the increased collaboration between different work groups, to a scientist's responsibility to disseminate the results of their research, and to complying with open access policies.

Open access provides transparency in the business model of the publication, but researchers do not always have the economic or technical support of their institution and on occasion research does not allow for its content to be openly shared (Fig 2). However, the value afforded to the reasons for openly publishing is higher than that for those hindering open access.

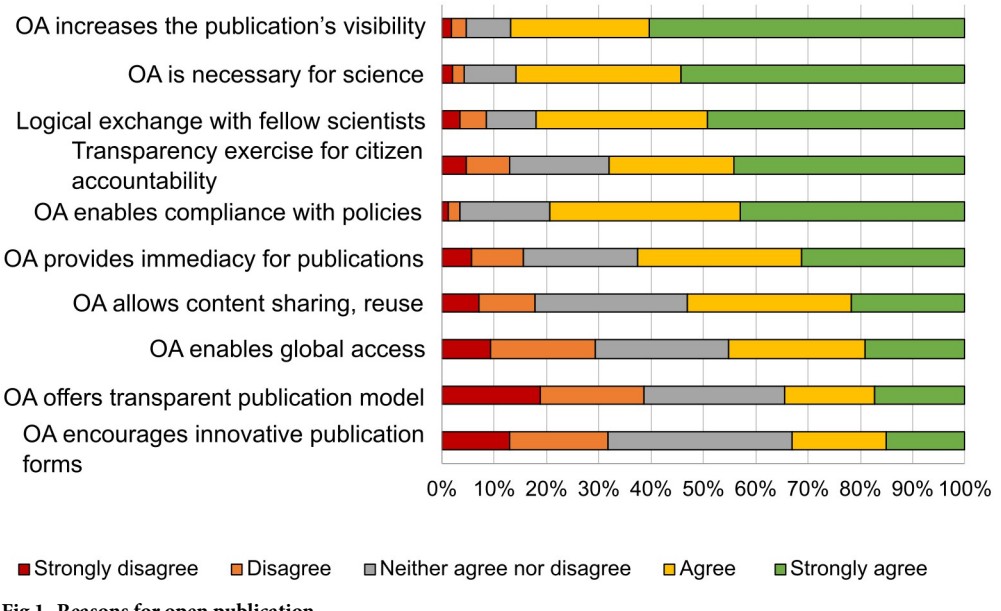

**Fig 1. Reasons for open publication.**

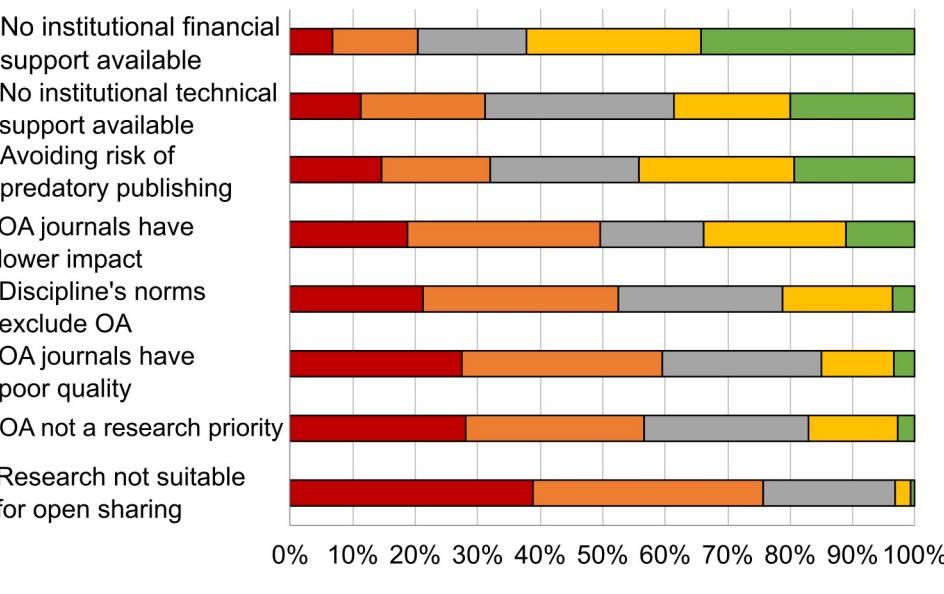

**Fig 2. Reasons for not being in agreement with openly publishing the results of research.**

Institutions, according to surveyed researchers, play an important role in increasing open access to scholarly outputs and supporting researchers, especially in terms of economic issues (APCs), but also in training and support for identifying and excluding predatory journals, for example. When comparing the scores for the reasons and hindrances to open publication, there are few effects dependent on discipline, age or gender of those surveyed ($\eta2 = 0.002$–$0.024$).

One of the very aspects highlighted by participants in the questionnaire is that of the funding problems that may arise when publishing in a commercial open access journal, because this favours inequality between teams with projects with funding and emerging research groups, as revealed in the survey comments.

> "Open publication only benefits the large groups who can easily access the knowledge of smaller ones and compete in unequal conditions. So, open publication, far from democratizing (I share my knowledge with colleagues I value, respect and trust), it increases inequality by excessively favouring large groups with a much greater capacity for competing since public funding always favours larger groups because they appear to have a safer value and be more profitable"

[R_46].

> "Open access is detrimental to emerging groups or countries with few economic opportunities in terms of disseminating innovative research. Open access only favours researchers with a lot of money, regardless of whether their ideas are innovative or will have an impact on society"

[R_77].

On the other hand, when researchers are asked about their preferred means of disseminating their research work with open access, the preferred venue was in an open-access journal (*N* = 303,47%), ahead of hybrid journals (*N* = 171, 26%) or storage in a repository (green way, *N* = 142, 22%). However, choosing one journal or another does not impede their ability to archive a copy in an institutional or discipline repository, having a licence to do so. With regard to pre-prints, only around 4% stated that it would be their preferred means, which gives interesting context to the myriad literature surrounding pre-prints as an alternative to the long publication times in many journals and which COVID-19 itself threw into question [28].

University libraries are key pillars to support researchers regarding scholarly publications. Open access has also posed a challenge for them when assuming roles they did not have some years ago, such as managing repositories, guaranteeing their quality, advising on editorial policies with regard to reuse licences, and so on. For this reason, interviewees were asked for their opinion on activities carried out by libraries in order to facilitate the publication of articles in open access. The results (Fig 3) show how total or partial funding for APC fees is the most valued aspect, however, support in terms of intellectual property matters and on editorial policies was also highly ranked, as well as open access publication and use of academic social networks.

When the question referred to support demanded from institutions (Fig 4), we observed that the main institutional support requested was funding for open publication (APCs or via transformative agreements), which allows for compliance with open publication requirements and publication in journals that are more relevant to their working fields.

In this case, participants in the questionnaire also consider that there should be greater support with regard to open access publication activities from their own institution. They find that support from their institutions is scarce, as mentioned in the following testimonies:

"There should be more training and information at an institutional level regarding open publication"

[R_98].

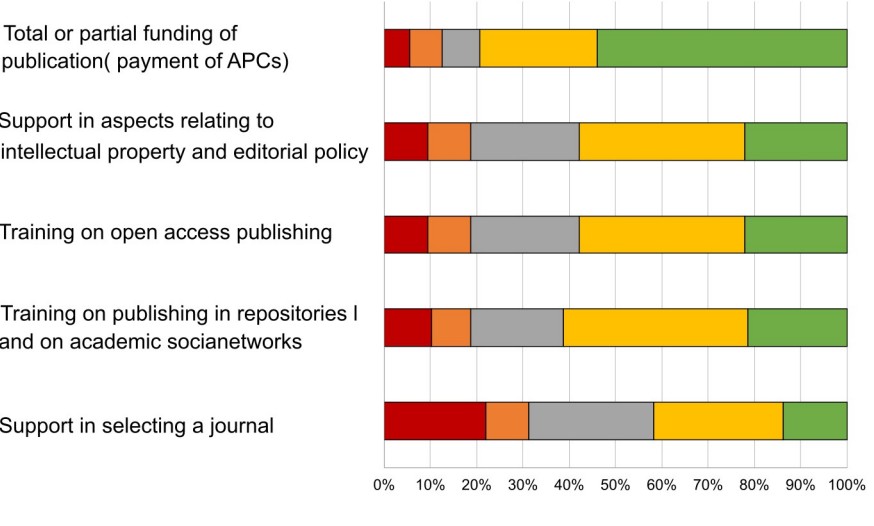

**Fig 3. Library support services about open access.**

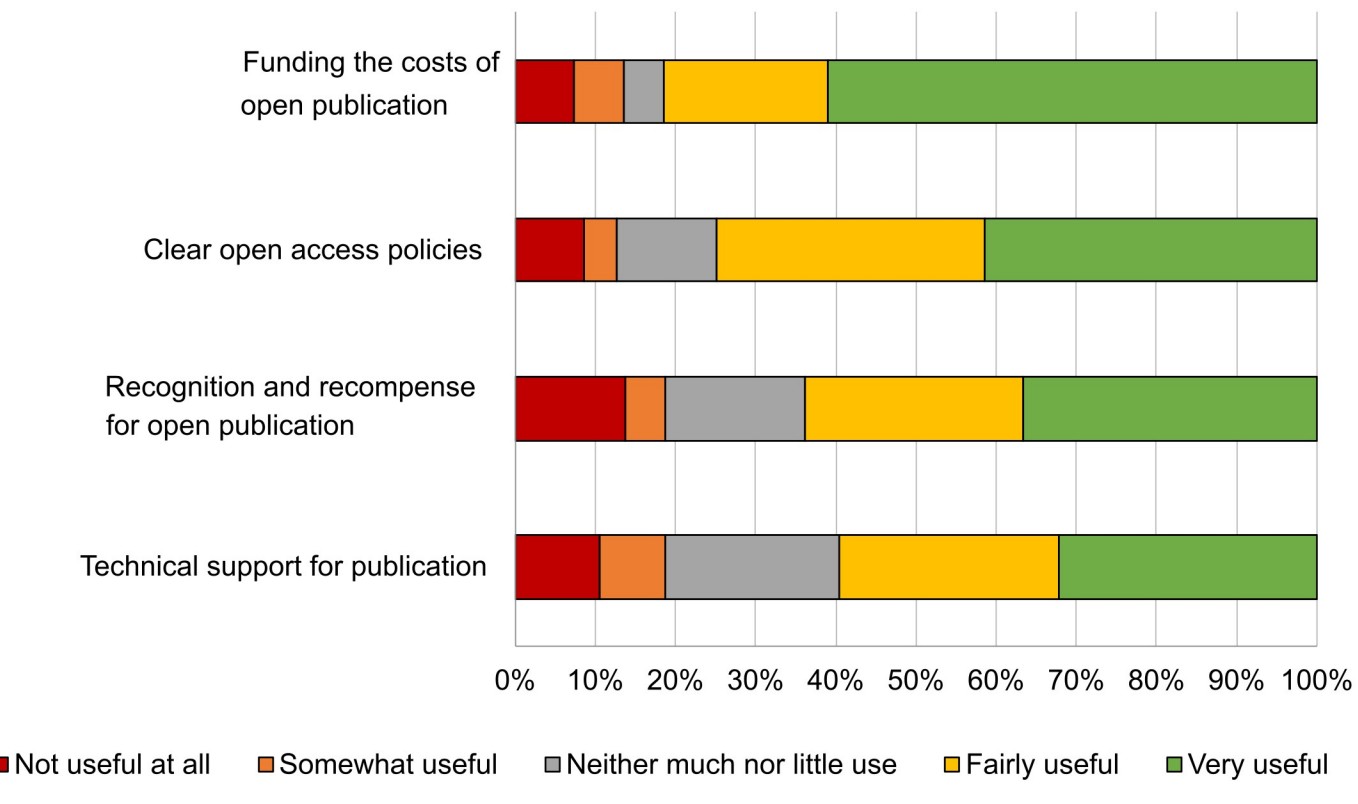

**Fig 4. Demands for institutional support.**

"My institution has no support activity on open access publication. All publication costs relate to projects, and often these costs are high and cannot be covered by the funding"

[R_120].

## 3.4 On the publication process, neither very free nor very open

Another key aspect of open access is the selection of the journal in which to publish and the relevance this has to researchers. In the responses (Fig 5), we see that indexation in databases and quartiles occupied by the journals occupied the first place, followed by the journal's specialization by discipline. Much less importance is given to whether it is an open access journal or whether copyright is retained by authors or not.

Publication time, which would be important for the quick dissemination of results, seems to be a lesser factor than that of the quartile occupied by the journal.

On the opinion of researchers about APCs, 70% of respondents stated they had paid APCs in the last two years. In terms of their opinions on the cost of this, as seen in Fig 6, most responses stated that they are generally excessive or unnecessary, although they understand that said cost relates to the journal's quartile.

In general, participants expressed in their questionnaires that APCs are excessive and often do not align with the quality of the journals, which for some is defined by the quartiles themselves.

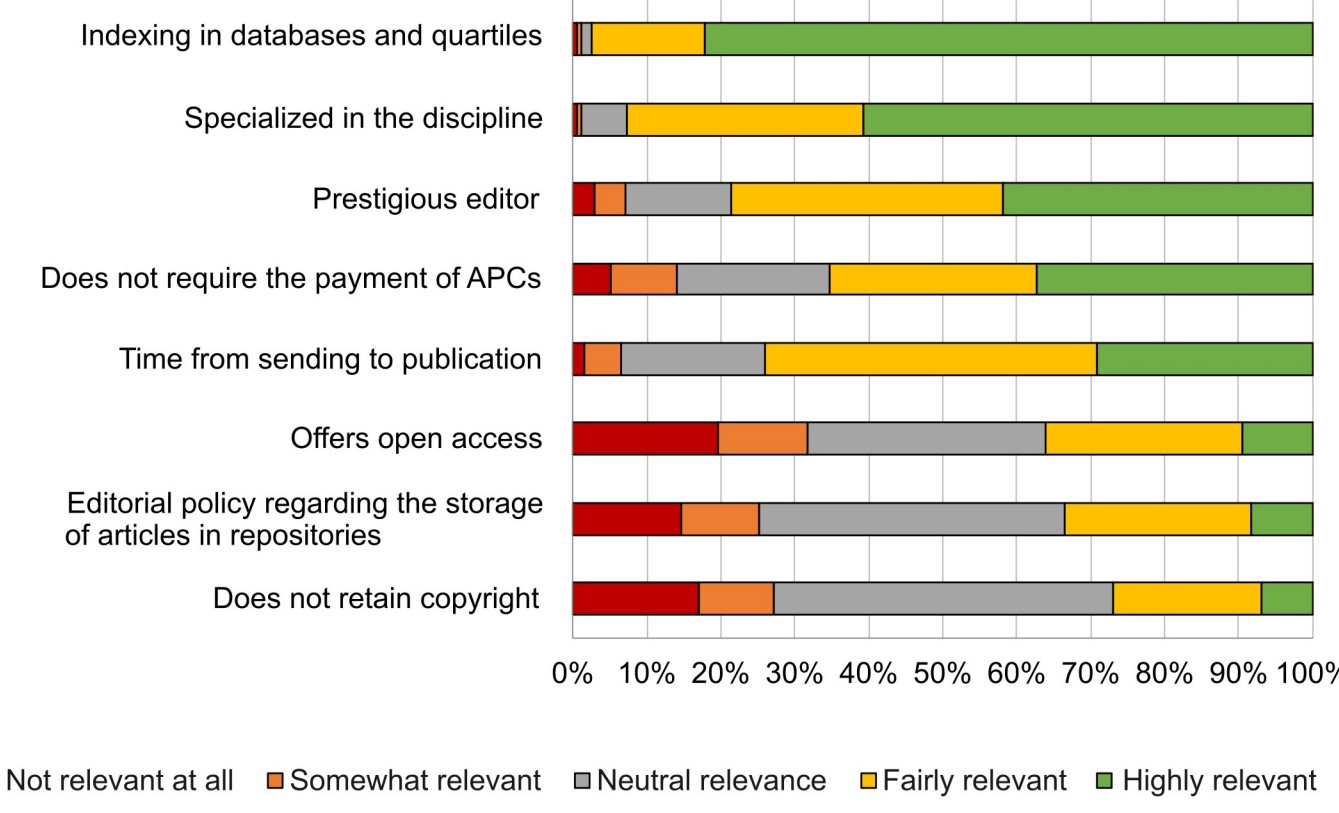

**Fig 5. Relevant aspects in terms of selecting a journal in which to publish.**

"In general, I think the payment of APCs does not closely correspond to a journal's quartile. There are several Q2s and Q3s with APCs and some Q1s without APCs. Ultimately, if I can pay I always look for a Q1 (also because it is what accrediting entities in Spain look at, not because I believe that a Q1 is better than a Q2 simply because an article is in one quartile or

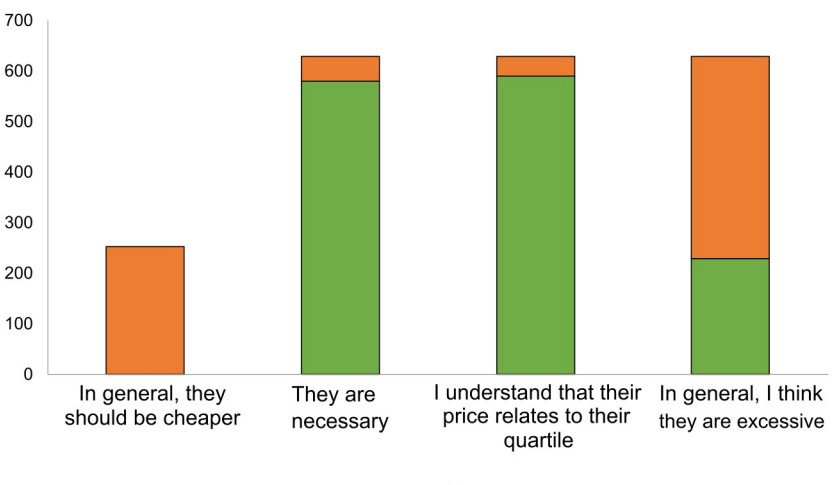

**Fig 6. Opinions on APCs.**

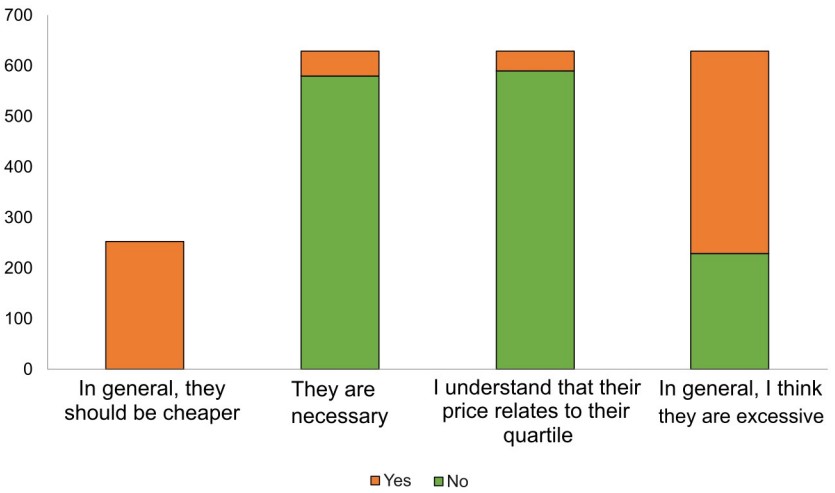

**Fig 7. Funding of APCs.**

another). If I don't have funding, I look for a Q1 without APCs, and if I can't find one that fits, I go to other quartiles"

[R_67].

When asked about where funding for the APCs came from, the majority of those surveyed answered that it came from the project or research group, or to a lesser extent from aid funding from the university/library (Fig 7). In very few cases, they referred to personal funds, which highlights the precariousness of funds dedicated to research in some sectors.

To conclude this block, interviewees were asked for their reasons for paying for APCs (Fig 8).

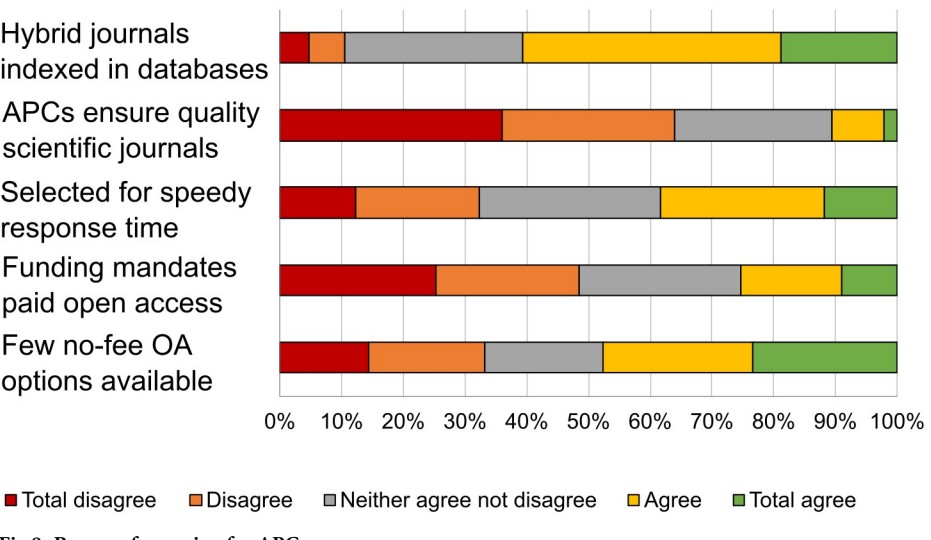

**Fig 8. Reasons for paying for APCs.**

### 3.5 Open research data, more support and more training

After open access to scientific publications, the sharing of research data is one of the main aspects regarding open science policies (the science act and royal decree 24/2021) and even within European directives on the reuse of data (directive 2019/1024 of the European Parliament and of the Council of 20 June 2019), relating to open data and to the reuse of information in the public sector) [29].

The first few questions of the 10 from the block on research data related to the funder and institutional policies for facilitating and promoting open access to research data in the scientific community. As shown by Fig 9, approximately one third stated they did not know whether their funder or institution had policies requiring research data sharing or requesting data to comply with FAIR principles, and a minority pointed out that they were aware of this but that they were exempt from the need to comply due to the nature of the data, following the mantra "as open as possible, as closed as necessary".

In answer to the question regarding whether they had openly shared their research data, those who had not and those who had, constituted around 43% each, while that percentage increased with those who intended to do so in the near future (14.5%).

Those researchers who responded affirmatively to the above question were asked where they had deposited their data. The responses are shown in Table 3, where we can see the range in the selection of repositories.

Those who stated they had archived research data in a repository (N = 240) indicated that their main reasons for doing so were the transparency and visibility of data and the ability to validate the results (Fig 10).

However, those who stated they had never shared data indicated that their reasons for not having done so (Fig 11) were related to matters of privacy and confidentiality, the lack of mandates a or their own lack of knowledge of how to do so; these reasons were also in agreement with other cases documented in the literature [30].

In the questionnaire, participants generally indicated that research data are costly to obtain, for which reason it should be available at the request of other authors rather than open to the general public.

> "Data compilation is a long and costly process, and I don't think it should be open to the general public for free. I don't think they should pay financially for it, but there should be justification for sharing it. Whenever I can, I use the option of sharing data at the explicit

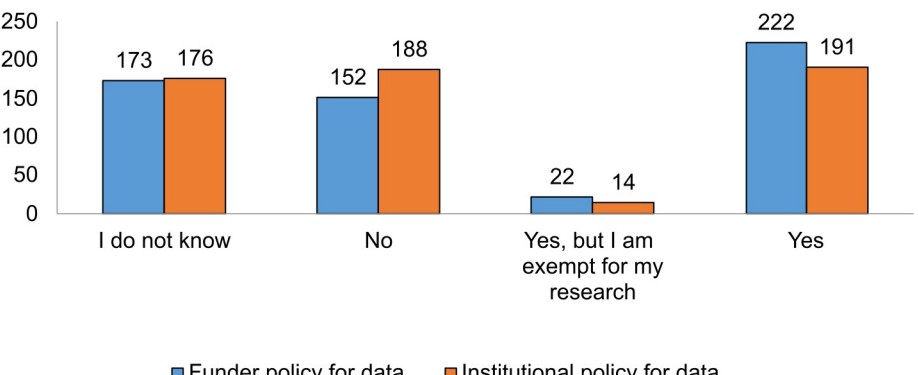

**Fig 9. Open access data policies funder and institution.**

**Table 3. Range of repository storage.**

| Type of repositories for self-archiving | N | % |
|---|---|---|
| Institutional repository | 117 | 33.6% |
| Funding agency repository | 5 | 1.4% |
| Specialist repository | 38 | 10.9% |
| General repository (Zenodo, Figshare…) | 52 | 14.9% |
| Research data repository (Dryad, Genbank, Dataverse…) | 85 | 24.4% |
| Data journals | 51 | 14.7% |

request of the authors, organizations and institutions that require it. There should be ethical reasons or relevant research matters (such as meta-analysis) for sharing data, otherwise it opens the door to a proliferation of secondary studies of low or dubious value in my field. We live in a model of scientific hyper-production, where quantity is prized over quality, and as much as possible I prefer not to facilitate ethically dubious behaviour that profits from the effort required to compile quality data"

[R_309].

Only 37% participants responded they had used their colleagues' data, and one of the main reasons for not using them (Fig 12) was a preference for reusing data the results of which have been published before.

In terms of support facilitated by the institution for the storage of research data, 77% of respondents stated that there should be more technical support for data management as well as for training in topics relating to it: training in open data management 65%, anonymization 51%, standards, metadata and IDs 56%, data management plans 50%.

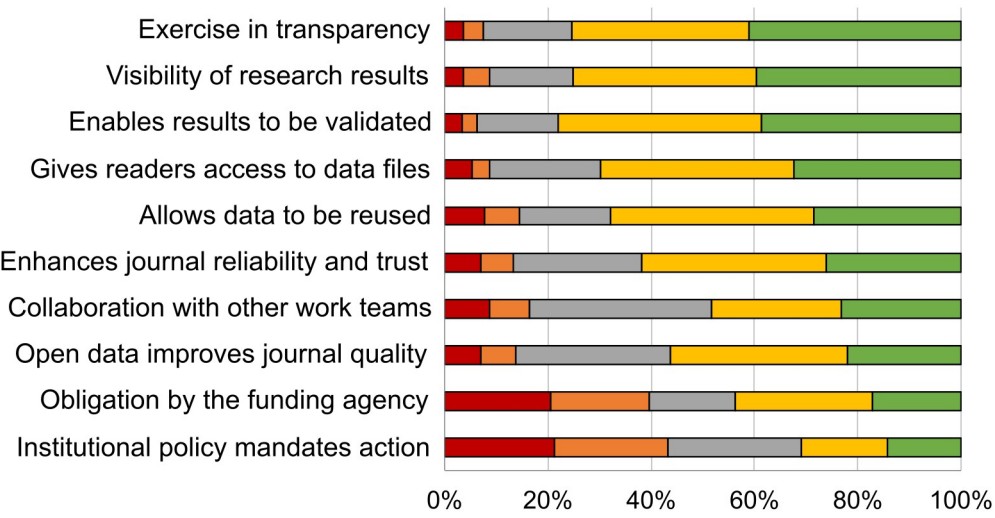

**Fig 10. Agreement on the reasons for which research data was stored.**

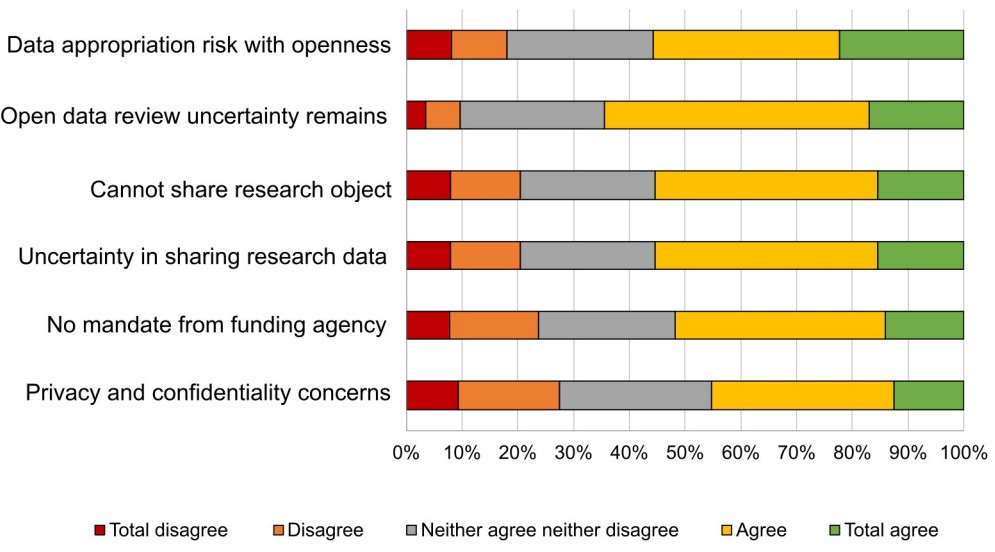

**Fig 11. Reasons for not storing research data.**

When looking at the top three circumstances that would incentivize them to share their data, the main responses were: quotation of their research work (67%), greater impact and visibility of their papers (61%) and some type of benefit or mandate from the journal/editor (together 56%). In addition, when asked who they would be prepared to receive support from, the higher-scored answer was that of editors (41%), closely followed by those within their own institution (38%).

Altogether, researchers' responses pointed to the need for training in research data management issues and how to comply with FAIR principles, and institutional support for facilitating data management tools. If the next 10 years can progress at the same rate as the last decade in terms of open research, the priorities should be: quality metadata, specific discipline metadata, and FAIR metadata.

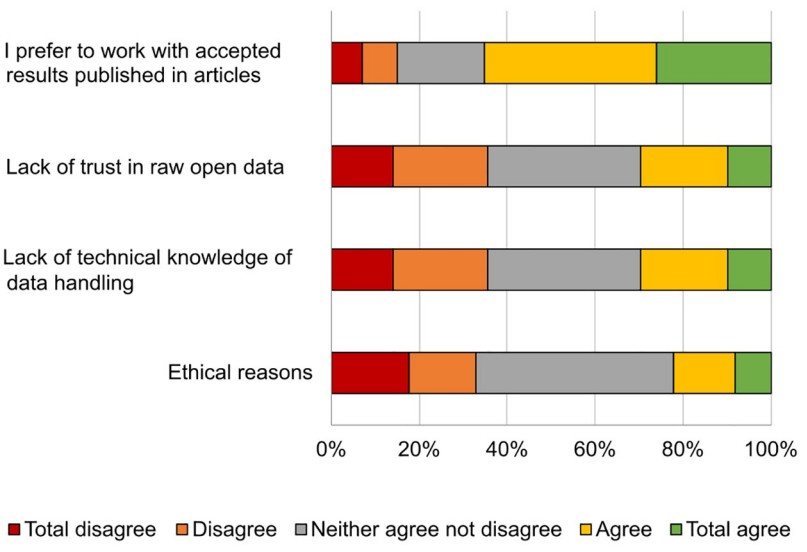

**Fig 12. Reasons for not reusing data.**

### 3.6 Open review, an incorporated practice, albeit flawed

The survey included four questions on open peer review (OPR) to find out if this modality of the peer review process, related to transparency [31] and designed to improve quality [32], was known to or practised by the academic community.

So, when asked if any of the journals in which they had published in the last two years (2019–2020) had an open peer review system, 52% of researchers responded affirmatively, and of them, 69% stated they had agreed for the review to be public, and 50.2% stated that they had performed reviews for journals that used this peer review system.

In terms of the researchers' opinions on open peer review (Fig 13), the anonymity of the reviews prevents any fear of their identity being discovered, particularly among young researchers, and that unveiling their identities could generate conflicts of interest between parties. In this sense, the results agreed with previous notes from studies such as that of Bjork [33].

With regard to this, participants in the questionnaire indicated objections to open review since it could generate conflicts of interest or because lower quality content might be accepted in order to avoid any confrontation. In addition, the hierarchical structure of senior versus junior researcher might affect reviews. In short, there is a tendency towards preferring traditional peer review and applying open peer review with a degree of caution.

> "In my opinion, peer review should be open; this would have enormous benefits for scientific debate, science would advance more rapidly and the quality of publications would be increased. The problem is that it can lead to misunderstandings or create conflict between research groups. It should be applied with caution"

[R_336].

> "This is a delicate topic, because there are pros and cons. In general, I agree that the more transparency the better, but in this case, the way of implementing this cultural change is also very important, and it would need to be done in an overall manner, with some rules and clear procedures in order to avoid conflict"

[R_125].

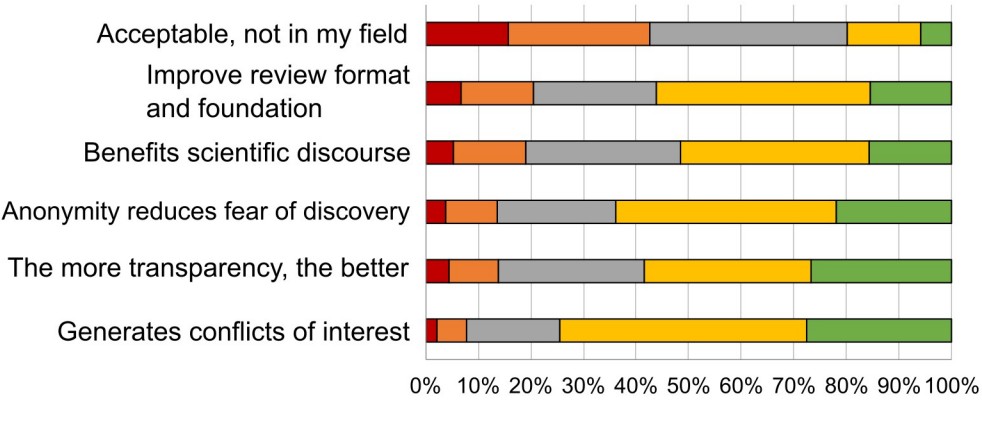

**Fig 13. Opinions on open peer review.**

### 3.7 Correlations between variables and groups of individuals

For the questions the responses to which were placed on a Likert scale, a factorial analysis was performed in order to reduce the number of variables and analyse their correlations. Table 4 shows the number of factors obtained for each of the blocks of questions together with the KMO value which indicates the robustness of the factorial analysis. The meaning of the factors relates to those variables with values of the original coefficient of the rotated component higher than 0.7. With this criterion, each of the blocks of questions was reduced to two factors, with the exception of the block referring to reasons for the non-use of third party data (1 factor) and to the non-storage of open data (3 factors).

As seen in the correlations between the various components, there is a negative correlation between the reasons for favouring open access and the usefulness and relevance of services offered by the library and institution, as well as relating to the advantages of open peer review.

**Table 4. Components obtained from each block of questions having applied a factorial analysis.**

| Question | Factors | Meaning | Correlation |
|---|---|---|---|
| Reasons for OA (KMO = 0.908) | FAC1_OA (OA1) | Compliance with policies, accountability, exchange of knowledge | |
| | FAC2_OA (OA2) | Transparency, innovation, immediacy, ability to be reused | |
| Reasons against OA (KMO = 0.751) | FAC1_NoOA (NoOA1) | No OA journals exist in my discipline, IF low, low quality, predatory risk | USEFULNESS1(-)**, USEFULNESS2(-)**, RELEVANCE2(+)**, OPR1(-)* |
| | FAC2_NoOA (NoOA2) | Lack of financial and technical support | OA1 (-)**, OA2(-)** |
| Usefulness of library and institution services (KMO = 0.856) | FAC1_USEFULNESS OF SERVICES (USEFULNESS1) | Support from library in the selection of journals, training in OA publication, training in academic networks | OA1(+)**, OA2(+)** |
| | FAC2_USEFULNESS OF SERVICES (USEFULNESS2) | Institutional support with payment of APCs, clear policies, institutional technical support, recognition and recompense pro OA | OA1(+)**, OA2(+)** |
| Relevance of the journal (KMO = 0.681) | FAC1_RELEVANCE (RELEVANCE1) | Choosing a journal because it allows self-archiving and there is no cession of rights | OA2**, USEFULNESS1(+)**, USEFULNESS2(+)* |
| | FAC2_RELEVANCE (RELEVANCE2) | Choosing a journal for its position in quartiles, because its IF and prestige | |
| Reasons for paying APCs (KMO = 0.605) | FAC1_PAYMENT APC (APC1) | Payment for quality and speed | |
| | FAC2_PAYMENT_APC (APC2) | Because of the discipline and because it is a requirement | |
| Reasons for data sharing (KMO = 0.845) | FAC1_REASONS DEPOSIT_ (DEPO_OD1) | Enables reuse, transparency, visibility, collaboration, access, trustworthiness, enables validation | OA1(+)**, OA2(+)**, RELEVANCE1(+)**, OPR1** |
| | FAC2_REASONS DEPOSIT (DEPO_OAD2) | Requirement of the funder | REASONS_ NO_USE_OAD1(+)** |
| Reasons against data sharing (KMO = 0.803) | FAC1_REASONS NO DEPOSIT (DEPO_NoOAD1) | Reticence/fear of competition, possible exploitation of results, avoidance of misuse, lack of knowledge on where to store data | RELEVANCE 1(+)* |
| | FAC2_REASONS NO_ DEPOSIT (DEPO_NoOAD2) | I cannot share it (closed) | USEFULNESS1(+)** |
| | FAC3_REaSONS_NO_DEPO (DEPO_NoOAD3) | My funder has no data policy | OA2(+)*, RELEVANCE2(+)* |
| Reasons against using third parties data (KMO = 0.694) | FAC1_REASONS NO USE (No USE_OAD1) | I do not know how to do it, for ethical reasons, due to mistrust, I prefer data which has undergone peer review | RELEVANCE2(+)**, DEPO_NoOAD1(+)**, DEPO_NoOAD2** |
| Opinions about OPR (KMO = 0.734) | FAC1_OPINION OPR (OPR1) | It favours transparency and fosters collaboration | OA1(+)*, OA2(+)**, USEFULNESS1(+)**, RELEVANCE1(+)** |
| | FAC2_OPINION OPR (OPR2) | It can create fear among young people and generate possible conflicts of interest | DEPO_NoOAD1(+)**, NoUSE_OAD1(+)** |

In contrast, the options in favour of OA positively correlate to the utility of services provided by the library and institution in training and support activities for facilitating open access to their publications. There is also a positive correlation between the value of OA and favouring of data sharing and the transparency conferred by open data.

### 3.8 Effect of COVID-19

In answer to the question of whether the COVID-19 pandemic had any effect on the rate of publication of articles, 35% of responses indicated that it had no significant effect (36%), 28% of researchers pointed out they were able to publish more due to the effect of lockdown, 22% said they had been affected by its health and social impact and 17% stated that they had published less due to the online teaching load.

Although the data from the questionnaire indicated that the pandemic did not have a significant effect on the load, participants also indicated difficulties in the continuity of their work and, in particular, difficulties reconciling family commitments with their regular work. They also indicated that they took advantage of the lockdown in order to publish outstanding works.

"The family-school-work balance was even more complicated, with less time (and lower quality time) for tasks requiring a high level of concentration and sustained attention such as the production of scientific articles"

[R_47].

"The months of lockdown during which no advances could be made in experimental work increases publications by allowing (or rather forcing) the dedication of time to half-finished work, pending data, and so on"

[R_103].

### 3.9 Final consideration and future trends

To conclude the survey, some questions were added in relation to the vision of the future of scholarly publication awareness of the principles enforced by Plan S. In this sense, we were surprised that 85% of respondents did not know of Plan S at the time of responding to the survey, although this may be logical if one bears in mind that no Spanish funding agency has adhered to Plan S. Also, among those who did know of it, there were those who approved of it (8.9%) and those who did not think it was a good idea (6%). These results can be placed in relation to the existing debate among the scientific community due to the reasons expressed in some open letters promoted by researchers in some disciplines in particular: the impossibility of finding journals in the upper quartiles that are 100% open access, and the loss of freedom in choosing where to publish.

With regard to the question of whether they would agree with a system where all articles were open and it was only necessary to pay to publish, 62% of those surveyed responded that they were against this, although approximately 62% believe that in the future, all articles will be open.

In this sense, there is some consensus among participants in indicating that the costs of open publishing should not fall so roundly to research groups.

"An agreed-upon model would be ideal so that open publishing is not so burdensome for research groups and/or the high cost of projects is affected"

[R_57].

The complex point, in relation to the above questions, is the need to pay to publish. In this sense, the future of Open Research Europe as a new alternative platform that should not compete for evaluation with high-impact factor journals seems to be an advantageous opportunity for innovating and changing the academic communication system. In fact, the very model and concept of the academic journal as it is currently known will probably change in the next few years. In this sense, 43% of those surveyed lamented the changing of the model, although 24% responded with a "no, I hope", implying a desire for change. As such, two thirds of the respondents, albeit for different reasons, think that the end of scientific journals is near, which is remarkable in such an inertial environment.

## 4. Discussion and conclusions

In 2011, a new licensing law was passed that incorporated an article on the open access to scientific output (Science Act 2011) which required the open access to scholarly publications, similar to the Royal Decree on open access to doctoral theses, passed in September 2011 (RD, 2011). In September 2022, the Spanish law (Ley 14/2011) was passed for Science, Technology and Innovation.

In any case, the effects of this legal framework have been largely irrelevant, as demonstrated by the percentage of Spanish researchers who disseminate in open access (40% of those surveyed do not publish any of their articles openly or less than half of their academic outputs, according to data from the survey). The main problem has been the lack, until recent years, of open access mandates and monitoring of open access driven by the main research funding programme, the R&D&I State Plan. From 2017 onwards, calls made specific reference to open access publications and in 2021 this was also extended to open research data.

Curiously, and in reference to open review, one of the elements of open science that has least been developed, it becomes evident that some Spanish researchers have published or reviewed in journals that offer the opportunity for open review. This data contrasts with the opinion of Spanish editors [22], who are highly reluctant to incorporate this model of review.

It is clear the scarce knowledge of the concept of open science (just 25% stated they knew what it was and that they agreed with its principles). In any case, as we have seen, researchers carry out open science practices conditioned by the requirements of calls to research or by the open review criteria of many international journals. There is no knowledge of the general framework, it is for this reason that it seems timelier than ever for there to be a greater effort at dissemination and training on open science on the part of universities and research centres, with particular emphasis on young researchers.

The recent revision of the aforementioned Science Act (2022) incorporated research data management and the possible use of open articles in the evaluation of research. The recently passed University System Ordinance Act (LOSU) consecrates its article 12 in the "Fostering of open science and Citizen Science", making reference not only to open access and FAIR data but also to citizen science and to the evaluation of research. In any case, these are generic statements in organic laws that have no direct translation into everyday life in the way that the approval of a state plan on open science might, such as those present in many European countries.

The overall conclusions are as follows:

- The meaning of open science and its implementation are emerging topics that still require promotion, training and even persuasion to be carried out.

- APCs present an economic barrier to those groups that do not have funding available for paying for them, although they are not the main criterion when selecting a journal.

- Although OPR practices may seem innovative and emerging, many researchers have published or reviewed in journals that offer the opportunity for open review.

- tenure and evaluation policies do seem to have a clear effect on the behaviour of researchers with regard to open science.

- Resolving the issue of evaluation in research may clearly condition the habitual uses of researchers when selecting a journal as well as the payment of APCs.

- Researchers state that they share research data more for reasons of persuasion than out of an obligation to carry it out.

- Researchers do not seem to question the objectives of open science, although they do question the pathways or difficulties that may arise from day to day.

- Open access to publications and open research data are two realities that are increasingly consolidated in the scientific community.

- Researchers seem aware that we are undergoing change, where academic evaluation or policies relating to open science, its implementation and shifting habits among researchers may change. In this sense, more and better support is needed on the part of institutions and teacher support services.

Conclusions linked to the publication process of scientific articles indicate that the position of a journal in quartiles and indexation and specialization in the discipline are the two main aspects when selecting a journal, ahead of sending time or the absence of APCs. The vast majority of scientists have published in the past two years in journals that included APCs, although they paradoxically believe that these are generally excessive or unnecessary or that there is no correlation between their price and the journal's quartile. Said APCs mostly came from project funding, although to a lesser extent they came from the research group's budget or institutional aid. Regarding the reasons for paying for APCs, researchers stated that most hybrid journals in their discipline are indexed in recognized databases or that there are almost no open journals in their discipline that do not require payment of APCs.

Regarding open access to research data, the main reasons for making it a reality relate to visibility, validation and transparency, which coincide with the objectives of open science and, to a lesser degree, to the obligation of the funder. The main reasons for not sharing research data relate to the non-obligation to do so and to aspects relating to privacy and confidentiality, followed by a lack of knowledge of how to do it. The vast majority of researchers believe that their institution should provide more support in advice and training regarding data management.

In terms of open peer review of scientific articles, most researchers responded that in the past two years (2019–2020) they had published in journals with an open review system and that they had agreed for the review to be public. At least 50% of respondents also stated that they had reviewed articles in journals that apply open peer review as an evaluation system by default.

Based on these conclusions, the authors indicate a series of recommendations to be considered by various interest groups relating to the research collective. Firstly, on the part of researchers, we reveal that researchers should systematically demand that their institutions provide technical, financial and training support; researchers should be ready for the turning point that may arise with the possible changes to evaluation policies in addition to informing themselves on new platforms, policies and review methods that may appear in the coming years.

In the field of scientific and information policies, the academic collective is aware that we are undergoing a change in such a way that it is possible to focus the debate and reflection surrounding research evaluation policies with the first point centred around the use of the impact factor. Researchers demand clear open access and open science policies which are accompanied by support and financial resources to facilitate new practices.

In university libraries, there should be systematized work with research collectives who do not publish openly and who feel unsure about predatory journals or who do not have sufficient knowledge of how to preserve confidentiality and privacy in the dissemination of research data.

To conclude, universities should open internal reflection processes regarding open science and the cultural shift it may bring about, among both young researchers and the rest of the collective, and it may be useful to advance in internal evaluation policies in measures and indications that do not exclusively depend on the impact factor, in order to prepare the ecosystem for future changes.

## Supporting information

**S1 File.**
(PDF)

## Acknowledgments

The authors would like to acknowledge Aurora González-Teruel and Carolina Andreu for their support in the project.

## Author Contributions

**Conceptualization:** Candela Ollé, Alexandre López-Borrull.

**Data curation:** Remedios Melero.

**Formal analysis:** Candela Ollé, Alexandre López-Borrull, Remedios Melero, Josep-Manuel Rodríguez-Gairín.

**Investigation:** Candela Ollé, Alexandre López-Borrull, Remedios Melero.

**Methodology:** Candela Ollé, Alexandre López-Borrull, Remedios Melero, Juan-José Boté-Vericad, Josep-Manuel Rodríguez-Gairín.

**Supervision:** Candela Ollé.

**Visualization:** Remedios Melero, Juan-José Boté-Vericad.

**Writing – original draft:** Candela Ollé, Alexandre López-Borrull.

**Writing – review & editing:** Candela Ollé, Alexandre López-Borrull, Juan-José Boté-Vericad, Josep-Manuel Rodríguez-Gairín, Ernest Abadal.

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
