## [Decision Letter · Decision Letter 0]

15 Mar 2023

PONE-D-23-01585Habits and perceptions regarding open science by researchers from Spanish InstitutionsPLOS ONE

Dear Dr. Ollé,

Thank you for submitting your manuscript to PLOS ONE. After careful consideration, we feel that it has merit but does not fully meet PLOS ONE’s publication criteria as it currently stands. Therefore, we invite you to submit a revised version of the manuscript that addresses the points raised during the review process.

We look forward to receiving your revised manuscript.

Kind regards,

Joana Sousa

Academic Editor

PLOS ONE

Journal Requirements:

4. Please include your tables as part of your main manuscript and remove the individual files. 

Reviewers' comments:

Reviewer's Responses to Questions

**Comments to the Author**

1. Is the manuscript technically sound, and do the data support the conclusions?

Reviewer #1: Partly

Reviewer #2: Yes

Reviewer #3: Yes

2. Has the statistical analysis been performed appropriately and rigorously? 

Reviewer #1: No

Reviewer #2: Yes

Reviewer #3: Yes

3. Have the authors made all data underlying the findings in their manuscript fully available?

Reviewer #1: Yes

Reviewer #2: No

Reviewer #3: No

4. Is the manuscript presented in an intelligible fashion and written in standard English?

Reviewer #1: Yes

Reviewer #2: Yes

Reviewer #3: Yes

5. Review Comments to the Author

Reviewer #1: Habits and perceptions regarding open science by researchers from Spanish Institutions

Manuscript No.: PONE-D-23-01585

This is a paper that describes the results of the online survey on open science in Spain indicated different perceptions of researchers. However, in its current version, I cannot support publishing this paper in the journal Plos One. The reasons are the following:

1. The abstract should include an underscore the scientific value added of paper, this section, it must structure in only one paragraph and include the main conclusions and implications.

2. In the introduction, the authors show a background and context of study. However, this section could explain in more detail: What is the new knowledge or value added of study? What is the research question or objectives? What makes the applied methodology suitable and superior in comparison to existing studies? What is the expected new insight gained by applying the method?, this section could include structure of manuscript and to improve literature review.

3. In the second section, authors describe methodology. This section needs to improve description of instrument with questions per category and the theoretical support, sample and strategies to analysis that guarantee robustness and reliability of the results.

4. Third section describes the main results; this section is adequate. However, the authors should explain significance level and statistics tests used.

4. Discussion must be improved including an analysis of results in comparison with literature strengths and shortcomings of this analysis and a short outlook on further research requirements and possible research extensions.

5. Conclusions are adequate.

I wish that these comments could help the author to improve the paper.

Reviewer #2: The manuscript (text, tables and figures), "Habits and perceptions regarding open science by researchers from Spanish Institutions", focuses on the situation of open science in Spain and the results of a survey of Spanish researchers in 2021.

Major revisions

The bibliographic review refers both to declarations (senso lato), standards and laws (UNESCO, BOAI, European Union, Spanish rules...) as well as to scientific articles. In both aspects it is suggested to deepen the search for works on the habits of researchers, in order to offer a broader discussion. I also suggest insisting on reviewing in the previous literature other relevant topics, as these: how erratic the forms of results dissemination continue to be; the extreme situation of this phenomenon in the case of research data given their intrinsic diversity, in comparison with open access… This makes the difference between the success of the openning the articles and the difficulties for the research data.

The methodology used is rigorous for the field being analysed, including a more sophisticated multi-factor analysis for the Likert scale questions. I only have one question: are all the questions asked in the questionnaire reflected in the results or are some not included in the study? If so, please explain why.

The results are perfectly described, although it is worth noting that, in the figures, the questions are cut off. If it is not possible to show them in full, it is suggested that the questionnaire was published. If they are in the text itself when the data are given, it is suggested that they be highlighted with underlining/bold/quotes or any other typographical means.

The Discussion and Conclusions section is a bit sparse. As I have said, it is imperative to expand the literature review in order to this section went beyond the survey results presented. A comparison of the results in Spain with the situation in other countries would enrich the conclusions offered by the work. One major issue that emerges from the survey responses, the evolution of open access towards the APC payment model, is treated rather lightly. Being a global concern that affects countries with less investment in research, such as Spain, and which is expected to last over time, it is necessary to work on the bibliography in greater depth, for which http://hdl.handle.net/1866/21676 or https://doi.org/10.5281/zenodo.4558704 are suggested. Other results concerning research data can be found at https://doi.org/10.1371/journal.pone.0021101 or even for Spain at https://doi.org/10.3390/data5020029.

Minor revisions

-- "Open Access data Polis funder and institution" in this sentence, I think should be policies the term polis

-- "In terms of researchers' opinions on open peer review (Figure 14)" I think it's figure 13

-- "article 12 in the "Fostering of open science and Civic Science"" I think it is more widely translated as citizen science.

*3. Have the authors made all data underlying the findings in their manuscript fully available?

Not mentioned

Reviewer #3: There are some issues with the plots. Some of the legends are not translated from Spanish. Many of the questions are trimmed, e.g.: Most hybrid journals in my discipline a... This is likely to be solved at editing stage. I find slightly confusing the fact that "Yes" is represented in orange (red?) and "No" in green in some of the questions. Again, maybe for the editor to decide.

It would be nice to compare these results with other countries with wider adoption of Plan S. Does it have any sort of influence in author's perceptions of OA? Maybe for a follow-up article.

I couldn't find table 4 ("Table 4 shows the number of factors obtained for each of the blocks of questions together with the KMO value which indicates the robustness of the factorial analysis"). It would be important to see it in order to validate the factorial analysis.

6. PLOS authors have the option to publish the peer review history of their article (what does this mean?). If published, this will include your full peer review and any attached files.

Reviewer #1: No

Reviewer #2: No

Reviewer #3: **Yes: **Sergio Ruiz-Perez

---

## [Author Response · Author response to Decision Letter 0]

26 Apr 2023

Dear Joana and reviewers, Thank you for the time you have spent reviewing the article and for your comments and

suggestions. We are going to respond in an orderly manner below to each reviewer and

each topic. We will also add them in the corresponding places within the manuscript.

Candela Ollé

---

## [Decision Letter · Decision Letter 1]

26 Jun 2023

Habits and perceptions regarding open science by researchers from Spanish Institutions

PONE-D-23-01585R1

Dear Dr. Candela Ollé,

We’re pleased to inform you that your manuscript has been judged scientifically suitable for publication and will be formally accepted for publication once it meets all outstanding technical requirements.

Kind regards,

Rut Lucas-Dominguez, PhD

Academic Editor

PLOS ONE

Additional Editor Comments (optional):

Authors achieved to improve the manuscript following the instructions of reviewers and it is adequate to publish in the journal considering the importance of the topic.

Reviewers' comments:

Reviewer's Responses to Questions

**Comments to the Author**

1. If the authors have adequately addressed your comments raised in a previous round of review and you feel that this manuscript is now acceptable for publication, you may indicate that here to bypass the “Comments to the Author” section, enter your conflict of interest statement in the “Confidential to Editor” section, and submit your "Accept" recommendation.

Reviewer #1: All comments have been addressed

Reviewer #2: All comments have been addressed

2. Is the manuscript technically sound, and do the data support the conclusions?

Reviewer #1: Yes

Reviewer #2: Yes

3. Has the statistical analysis been performed appropriately and rigorously? 

Reviewer #1: Yes

Reviewer #2: Yes

4. Have the authors made all data underlying the findings in their manuscript fully available?

Reviewer #1: Yes

Reviewer #2: Yes

5. Is the manuscript presented in an intelligible fashion and written in standard English?

Reviewer #1: Yes

Reviewer #2: Yes

6. Review Comments to the Author

Reviewer #1: Authors achieved to improve the manuscript and it is adequate to publish in the journal considering the importance of topic.

Reviewer #2: (No Response)

7. PLOS authors have the option to publish the peer review history of their article (what does this mean?). If published, this will include your full peer review and any attached files.

Reviewer #1: **Yes: **Clara Inés Pardo Martinez

Reviewer #2: No

---

## [Editor Report · Acceptance letter]

4 Jul 2023

PONE-D-23-01585R1 

Habits and perceptions regarding open science by researchers from Spanish Institutions 

Dear Dr. Ollé:

I'm pleased to inform you that your manuscript has been deemed suitable for publication in PLOS ONE. Congratulations! Your manuscript is now with our production department. 

Kind regards, 

on behalf of

Prof. Rut Lucas-Dominguez 

Academic Editor

PLOS ONE